# Exploring Captive Giant Panda Reproduction: Maternal and Offspring Factor Correlations from 324 Breeding Events

**DOI:** 10.3390/ani15081182

**Published:** 2025-04-21

**Authors:** Bo Luo, Bo Yang, Qiang Zhou, Guo Li, Yanwu Lai, Wen Zeng, Guiquan Zhang, Desheng Li, Liu Yang

**Affiliations:** China Conservation and Research Center for the Giant Panda, Key Laboratory of State Forestry and Grassland Administration on Conservation Biology of Rare Animals in the Giant Panda National Park, Chengdu 610051, China; boluo911@126.com (B.L.); yangb_198701@163.com (B.Y.); zqzq13881725851@sina.com (Q.Z.); pandaapple2025@163.com (G.L.); 18380247479@163.com (Y.L.); 18227566688@163.com (W.Z.); zguiquan@126.com (G.Z.); lidesheng_ccrcgp@163.com (D.L.)

**Keywords:** panda, reproduction, cub birth weight, maternal age, neonatal mortality

## Abstract

This study analyzed 324 breeding events (1998–2023) to evaluate how maternal traits—including maternal age, interbirth intervals, and gestational duration—shape reproductive outcome in captive giant pandas. We focused on cub birth weight, neonatal survival, and number of cubs per breeding event as key indicators of offspring viability. Results revealed that younger dams (5–7 years), advanced-age dams (≥20 years), and short interbirth intervals (≤1 year) were associated with higher neonatal mortality, while mid-length gestational durations (110–127 days) correlated with optimal cub survival. These findings inform evidence-based suggestions to refine captive breeding protocols, such as prioritizing maternal recovery periods and monitoring older dams. The study developed here could aid conservation efforts for other vulnerable species by linking maternal health to population sustainability.

## 1. Introduction

The giant panda (*Ailuropoda melanoleuca*) is an important species for global biodiversity conservation but faces challenges in terms of breeding, including the low survival rates of cubs in early growth stages [1,2,3,4,5,6]. A smooth process of breeding captive giant pandas is crucial for maintaining a steady increase in the overall population. By the end of 2023, the captive giant panda population in China had reached 728 individuals, achieving the milestone of a self-sustaining captive population [7]. In addition to population size, factors such as survival success and the health of panda cubs are important for evaluating the effectiveness of panda breeding efforts. Previous studies have shown that maternal and offspring factors such as maternal age, gestational duration, and cub birth weight of newborn pandas impact panda survival [8]. However, these studies were limited by small sample sizes [8,9]. The current study analyzed data from 324 breeding events of captive giant pandas from 1998–2023, including factors such as maternal age, gestational duration, and interbirth interval, and their effect on the number of cubs per breeding event, cub birth weight, and neonatal mortality ratio. We also analyzed the relationship between cub birth weight and the number of fetuses and neonatal mortality. This study provides abundant information on the largest dataset of panda breeding events published to date. Furthermore, this study provides scientific guidance for promoting the healthy development of captive giant panda populations and optimizing the level of breeding management.

## 2. Materials and Methods

### 2.1. Data Collection and Selection Criteria

Breeding event records (*n* = 324) were obtained from the China Conservation and Research Center for the Giant Panda’s (CCRCGP) studbook database (1998–2023). The CCRCGP is a national institution under China’s National Forestry and Grassland Administration, oversees the world’s largest captive panda population, and leads research on breeding, feeding, and disease control [10]. Inclusion criteria required complete datasets for maternal age, gestational duration, cub birth weight, and neonatal survival status. The definitions of the factors are as follows:

Maternal age: age (years) at conception, calculated from studbook birth dates.

Interbirth interval: The time elapsed between two successive births (years). Zero represents the first time a female panda gave birth. ≥4 represents an interbirth interval of 4 years or more due to limited sample sizes beyond 4 years (n = 39) to ensure statistical power while avoiding overfragmentation.

Gestational duration: Duration was measured from confirmed fertilization (via sustained progesterone elevation) to parturition. This spans two biologically distinct phases: embryonic diapause and active gestation.

Cub sex: this was determined based on the appearance of the panda cub’s genitalia.

Cub birth weight: panda cubs were weighed via electronic scales within 0–2 h after birth, with an accuracy of 0.1 g.

Neonatal mortality: death within 3 months of birth or a stillbirth, which were verified against veterinary necropsy reports.

Neonatal mortality ratio: the proportion of cub deaths (stillbirths + deaths within 90 days) relative to total breeding events per dam.

### 2.2. Female Reproductive History

All females were classified as “healthy” based on monthly veterinary assessments confirming the absence of reproductive pathologies (e.g., endometriosis and ovarian cysts) or chronic systemic diseases (e.g., diabetes). All high-parity females (≥5 pregnancies) underwent annual reproductive health assessments, with veterinary clearance required for subsequent breeding.

### 2.3. Breeding Management Consistency

All breeding events complied with the national standard of the China Conservation and Research Center for the Giant Panda. Veterinary teams monitored maternal health identically across natural mating and assisted reproduction events, including: (1) daily urinary estrogen/progesterone assays during estrus; (2) pre-breeding physical examinations; (3) postpartum neonatal care protocols. All primiparous females met strict pre-breeding health criteria: (1) body condition assessed via rib palpation and abdominal contour; (2) no evidence of nutritional deficits in biannual blood panels.

### 2.4. Data Analyzed

All statistical analyses were conducted using SPSS (version 22.0) (Chicago, IL, USA). Continuous variables (e.g., cub birth weight and gestational duration) were reported as mean ± standard deviation (SD) or median (range) based on normality testing via the Kolmogorov–Smirnov test (*p* < 0.05 indicating non-normal distributions). Categorical variables (maternal age groups, interbirth intervals, and number of cubs per breeding event) were summarized as frequencies and percentages. Group comparisons were performed using ANOVA with Bonferroni post-hoc corrections for normally distributed data (e.g., cub birth weights across maternal age groups and gestational duration intervals) and the Wilcoxon rank-sum test for non-parametric data (e.g., birth weight differences between singletons and multiples and twins with/without mortality). Relationships between variables were assessed using Spearman’s rank correlation for non-linear associations (e.g., maternal age and litter size; gestational duration and cub birth weight). In addition, when the analysis involved birth years, birth years were aggregated into three-year intervals (1998–2000, 2001–2003, …, 2021–2023) to reduce bias from annual fluctuations in management practices. Detailed statistical frameworks including model specifications, validation protocols, and sensitivity analyses are comprehensively documented in the Appendix A.

## 3. Results

### 3.1. Distribution of Maternal Age, Cub Birth Weight, Gestational Duration, Number of Pregnancies, and Interbirth Intervals

Females (n = 91 total) ranged from primiparous to multiparous (1–9 previous births). Among them, 60 females (65.9%) contributed ≥ 2 breeding events to the dataset. Maternal age at conception ranged from 5 to 23 years across 324 breeding events, with a mean (±SD) age of 11.3 ± 4.1 years and a mode of 11 years (peak frequency: 12.0%). Notably, 54.4% of births occurred between ages 8 and 14 years (Figure 1A). Advanced maternal age (>18 years) accounted for only 6.8% of births, with the oldest dam recorded at 23 years (frequency: 0.3%) (Figure 1A, Appendix A). Cub birth weights ranged from 53.5 g to 245.0 g, with a distinct concentration in the 140–180 g range (51.5% of births). When grouped into 10 g intervals (>a–≤b), the distribution revealed two primary peaks: (1) 150–160 g: 14.5% of births (n = 59); (2) 170–180 g: 12.6% (n = 51). Notably, 91.1% of weights fell between 100 and 210 g, with extreme values (<100 g or >210 g) accounting for only 9% (Figure 1B, Appendix A). In addition, the cub birth weights of the 406 births did not follow a normal distribution according to the Kolmogorov-Smirnov (K-S) test (n = 406, *p* < 0.005). Among the 406 cubs analyzed, 209 were female and 197 males, with no significant sex difference in birth weight (Kruskal–Wallis U test: D = 1, *p* > 0.05). The mean cub birth weight ± standard deviation among female newborns was 156.57 ± 34.24 g, whereas it was 159.05 ± 31.09 g among male newborns. Cub birth weights showed no significant variation across nine three-year intervals (Appendix A; Figure 1C). The gestational duration averaged 125.06 days (range as 71–188 days) (Figure 1D), with a mode of 117 days (19 events, 4% of total), and exhibited a non-normal distribution (K-S test, n = 406, *p* < 0.005). The majority of females (78.1%) had 1–4 pregnancies, with the highest proportion being first-time pregnancies (34.1%). A sharp decline is observed for higher pregnancy counts: only 21.9% of females had ≥5 pregnancies, and this subset exhibited a progressively decreasing trend: 9.9% for 5 pregnancies, 3.3% each for 6–8 pregnancies, and 2.2% for 9 pregnancies (Appendix A). High-parity females (≥5 pregnancies, n = 20) showed comparable cub outcomes to low-parity counterparts (birth weight: 154.01 ± 33.45 g (≥5) vs. 169.74 ± 31.88 g, t = 2.065, df = 404, *p* < 0.05 (unpaired *t*-test)) (Appendix A). The interbirth intervals among all breeding events ranged from 0 to 4 years (median = 2). The majority of interbirth intervals (81.5%) fell within 0–2 years (Appendix A).

After excluding births where sex determination was precluded by neonatal mortality, 464 births were analyzed, comprising 234 females (50.4%) and 230 males (49.6%). The most common number of fetuses per breeding event was one (occurring 168 times, 51.9%). A total of 152 breeding events (46.9%) yielded twins, while only 4 breeding events yielded triplets (1.2%).

### 3.2. Relationships Between Maternal Age and the Number of Cubs per Breeding Event, Cub Birth Weight, and Neonatal Mortality Ratio

Spearman correlation analysis revealed a weak negative relationship between maternal age and the number of cubs per breeding event (β = −0.403, R^2^ = 0.162, *p* < 0.05). The bar plot illustrates maternal age-related trends in twin birth ratios (Figure 2A). Twin births peaked at ages 5–6 (66.7%) and 8 (68.8%), declined to a low at age 15 (8.3%), then partially rebounded by age 20 (60.0%). ANOVA analysis indicated a nonsignificant negative correlation between maternal age and cub birth weight (*p* > 0.05, details in Appendix A) (Figure 2B). Neonatal mortality ratios peaked at 20 years (25%, n = 12) and 13 years (21%, n = 29), while dipping at mid-ages (8 years: 3% (n = 37); 18 years: 0% (n = 10); 21 years: 0% (n = 2)). Mortality in younger females (5–7 years) was elevated, whereas advanced maternal age (≥20 years) showed the highest rates, though limited by small sample sizes (Figure 2C).

### 3.3. Relationships Between Gestational Duration and the Number of Cubs per Breeding Event, Cub Birth Weight, and Neonatal Mortality Ratio

No significant difference was revealed in gestational duration between singletons and twins (*p* > 0.05; Figure 3A). When categorized into six descending intervals (71–188 days), twin births dominated in mid-to-late gestation (92–162 days), with the highest proportion at 165–188 days, while only singletons occurred in the shortest interval (71–88 days; Figure 3B). Neonatal mortalities showed no significant correlation between gestational duration and cub birth weight (*p* > 0.05, details in Appendix A, Figure 3C). Neonatal mortality was lowest at intermediate durations (110–127 days; Figure 3D).

### 3.4. Relationships Between the Interbirth Interval and the Number of Cubs per Breeding Event, Cub Birth Weight, and Neonatal Mortality Ratio

The proportion of twin births was slightly higher in primiparous dams (0) and those with ≥4-year intervals (Figure 4A). ANOVA with Bonferroni-corrected pairwise comparisons revealed no significant birth weight differences among interbirth interval groups. A non-significant trend toward lower weights was observed in the ≥4-year interval group compared to the 1-year interval group (details in Appendix A, *p* > 0.05, Figure 4B). Neonatal mortality ratios were highest at 1-year intervals, intermediate at 2–3 years, and lowest at 0 and ≥4 years (Figure 4C).

### 3.5. Relationships Between the Number of Cubs per Breeding Event and Cub Birth Weight, and Neonatal Mortality Ratio

After excluding neonatal mortality cases, singleton offspring showed significantly higher birth weights than multiples (*p* < 0.01, Figure 5A). After excluding twin breeding events with incomplete weight records due to stillbirths, 133 events were analyzed, including 22 pairs with neonatal mortality. The weight difference in these pairs ranged from 0.3 to 175.6 g, and significantly greater weight differences in pairs with neonatal mortality were shown compared to those without (*p* < 0.05, Figure 5B).

## 4. Discussion

The distribution of pregnancy frequencies among female individuals underscores the limited representation of high-parity individuals, reflecting stringent management protocols to prioritize maternal welfare and ethical breeding practices in captive populations. Body weight serves as a critical indicator for assessing animal growth, development, and health status [11,12,13]. Analyzing the weight of newborn pandas enables the monitoring of pandas’ developmental and survival potential [14,15,16]. The distribution of cub birth weight in this study was similar to the weight distribution of 219 pandas in a previous study [14] as well as the weight distribution of 145 pandas in another study [17]. The differences in the average cub birth weight reported by Zhang [14], Huang et al. [17], and the current study were less than 2.23–13.17 g, thus confirming the reliability of the current study. No significant differences in cub birth weights were observed from 1998 to 2023, suggesting cub birth weight was not influenced by external factors or was not easily affected. But singleton offspring showed significantly higher birth weights than multiples in this study. Panda cubs with higher birth weights exhibit greater early mobility, enhancing their ability to compete for maternal milk and accelerate growth [18,19]. Zhou et al. found that the survival time of singleton newborn giant pandas was 1.88 times longer than that of twin newborns and 4.85 times longer than that of triplet newborns [8]. These findings indicate that a panda’s birth weight has potential implications for its survival and health. In addition, male cubs exhibited slightly higher mean birth weights than females [14,17]. Zhang et al. noted that male pandas significantly outweighed females from sub-adult to adult stages [14]. In wild giant panda populations, males with robust body conditions typically exhibit higher mating success due to their competitive advantage in securing access to receptive females [20]. This may indicate a normal phenomenon of environmental adaptation and natural selection for cub birth weight for the sex selection of giant pandas.

The variability in total gestational duration (71–188 days) primarily reflects individual differences in embryonic diapause—a facultative adaptation allowing delayed implantation until optimal maternal conditions are met [21]. In giant pandas, embryonic diapause is hormonally regulated, with progesterone levels maintaining blastocyst dormancy until metabolic cues trigger implantation [22]. While the diapause phase can extend for weeks or months, active post-implantation gestation is relatively constant, suggesting that total gestational duration variability arises predominantly from diapause length. This phenomenon complicates the interpretation of gestational duration as a continuous variable as prolonged diapause may not directly impact fetal growth. Our finding of no significant correlation between total gestational duration and cub birth weight supports the hypothesis that birth weight is governed by maternal nutrient allocation during active gestation rather than diapause duration [23,24,25]. However, the inability to retrospectively partition diapause and active phases in our dataset highlights the need for prospective studies combining ultrasonography and hormonal profiling to disentangle these effects.

In this study, singleton births accounted for the majority (51.9%) of all breeding events, followed by twin births (46.9%) and triplet births (1.2%). This distribution of the number of cubs per breeding event aligned closely with findings reported in previous studies [2,26], thus confirming the reliability of the current study. Furthermore, a significant negative correlation was observed between the maternal age of female pandas and the number of cubs per breeding event. This correlation may be attributed to the physiological status of female pandas at different ages. In previous fecundity studies of other animals, younger dams typically exhibit greater reproductive performance [27,28,29,30]. Moreover, the interbirth interval was found to influence the number of cubs per breeding event and cub birth weight. When panda dams gave birth for the first time or after a four-year interval, the proportion of twin births was slightly greater than the proportion of singleton births. Although there have been no specific studies on interbirth interval and maternal age in pandas, interbirth interval and maternal age are both associated with the physiological status of women [31,32,33,34]. During the shortest gestational duration range (71–88 days), the probability of giving birth to a singleton cub was highest; during the moderate-length gestational duration (92–162 days), the proportions of singleton births and twin births were roughly equal; and during the longest gestational duration range (165–188 days), the probability of giving birth to multiple cubs reached its peak. These results indicated that in the breeding process, maternal age, interbirth interval, and gestational duration are crucial predictors of the number of offspring.

Elevated mortality in younger dams (5–7 years) may reflect primiparous inexperience or physiological immaturity. Conversely, the unexpected mortality peak at 13 years—a prime reproductive age—hints at undiagnosed age-specific risks, such as hormonal dysregulation or cumulative reproductive fatigue, warranting targeted health screenings. The notably high mortality in advanced age groups (≥20 years) aligns with senescence-driven declines in placental efficiency or immune function observed in managed wildlife [35,36], yet small sample sizes (n = 12 at 20 years; n = 6 at 22 years) urge caution. This limitation underscores a critical gap in captive panda demography: few individuals survive beyond 20 years, and those that do are often selectively bred, confounding age effects with managerial practices. Furthermore, the more the gestational duration deviated from the median range (110–127 days), the higher the neonatal mortality ratio. Neonatal mortality ratios were higher with a one-year interbirth interval, likely due to insufficient time for the mother’s reproductive system and energy to recover between breeding periods [37,38]. In twin pairs where one cub died, the birth weight difference was significantly greater than in pairs where both survived, likely due to imbalanced fetal development affecting survival capabilities [39,40]. These results reflected the potential effects of maternal age, gestational duration, interbirth interval, and cub birth weight on the survival of panda cubs.

Pseudopregnancy, a common phenomenon in captive giant pandas, was excluded from this analysis to focus on factors influencing successful cub outcomes. Pseudopregnancies are characterized by progesterone profiles and maternal behaviors resembling true pregnancy without live birth [21,41]. While these cases highlight challenges in early gestation (e.g., embryonic resorption), our exclusion criteria ensured that gestational duration and interbirth interval calculations reflected biologically viable pregnancies. Future work comparing pseudopregnant and pregnant individuals could elucidate metabolic or hormonal thresholds critical for implantation success.

While our findings reveal significant correlations between maternal factors (age, interbirth intervals, and gestational duration) and cub outcomes, we emphasize the need for cautious interpretation due to four key limitations. First, as an observational study, we could not control for unmeasured confounders such as genetic diversity, paternal contributions, or keeper expertise. For instance, the association between shorter interbirth intervals and higher neonatal mortality may reflect institutional practices of preferentially breeding high-performing dams rather than inherent biological risks. Second, historical data gaps—notably maternal body weight at first pregnancy—constrain our ability to assess energy reserve effects, though age served as a stable proxy under managed feeding regimes. Third, the lack of paternal data precludes evaluating genetic influences on cub viability. Fourth, while pooling intervals ≥ 4 years improved statistical power, it may obscure potential differences between 4-year and longer intervals. Future studies with larger samples should explore finer categorizations.

Current giant panda breeding programs prioritize cub health over artificial manipulation of litter size or sex. Although this study analyzed the effects of gestational duration, maternal age, and interbirth interval on panda newborns, it does not aim to guide intentional manipulation for large fetuses or specific sexes per breeding event. Instead, this study aims to provide a reference for giant panda captive breeding by examining factors related to panda dams. Future research on giant panda reproduction should focus on physiological mechanisms, including oocyte maturation, follicular development, delayed implantation, and fetal development, which require a deeper understanding of their reproductive biology. While focused on pandas, the findings may provide a framework for studying female reproduction in other species.

## 5. Conclusions

This comprehensive analysis of 324 captive giant panda breeding events (1998–2023) elucidates critical maternal and gestational factors influencing offspring outcomes. Key findings demonstrate that maternal age significantly impacts cub survival metrics. Gestational duration and interbirth intervals further modulate reproductive success. The study underscores the importance of optimizing breeding strategies by prioritizing maternal health monitoring in giant pandas, particularly for aging dams, and allowing adequate recovery intervals between pregnancies. While no intentional manipulation of litter size or sex ratios is advocated, these insights provide transferable suggestions (e.g., optimal interbirth intervals and maternal age thresholds) for hypothesis-driven refinements of captive breeding protocols to enhance cub viability. These guidelines, adaptable to species-specific life histories, support broader biodiversity preservation efforts. By bridging empirical patterns with practical management, this work advances hypothesis generation for giant panda conservation while underscoring the need for experimental validation to establish causal efficacy.

## Figures and Tables

**Figure 1 animals-15-01182-f001:**
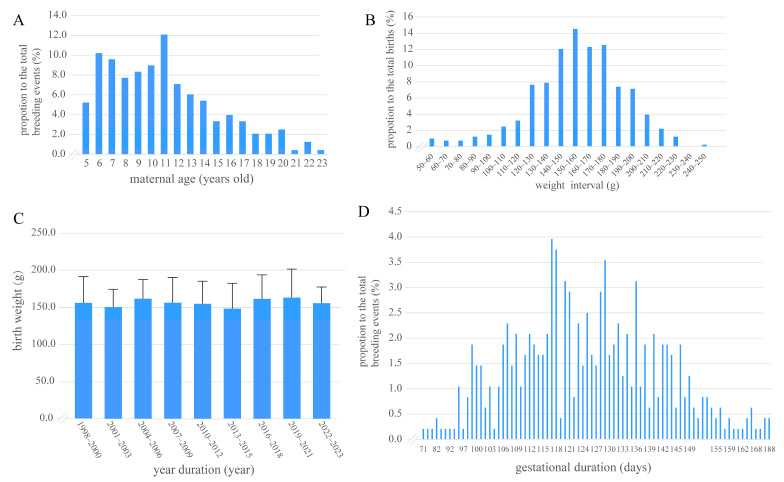
Distributions of giant panda maternal age (**A**), cub birth weight in 10 g intervals (>a–≤b) (**B**), and gestational duration (**D**) as percentages, and mean birth weight of cubs grouped by year (**C**). ((**A**) Distribution of maternal ages, n = 324 breeding events; (**B**) distribution of cub birth weight in 10 g intervals (>a–≤b). n = 406 births; (**C**) mean cub birth weight for nine different years groups, n = 406 births, error bars represent ± 1 SD using ANOVA with post-hoc Bonferroni corrections, *p* > 0.05; (**D**) distribution of gestational durations, n = 324 breeding events).

**Figure 2 animals-15-01182-f002:**
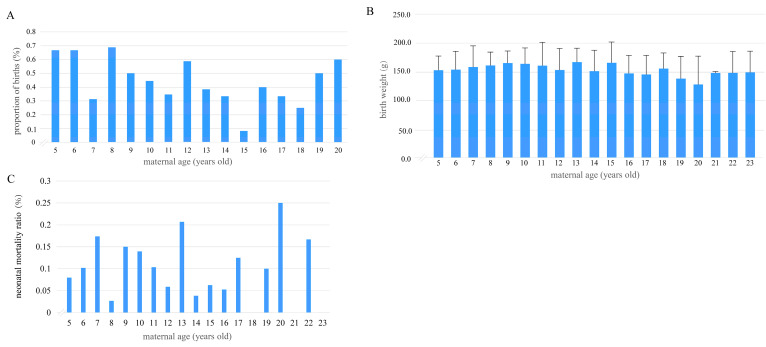
Relationships between giant panda maternal age and twinning rate (**A**), cub birth weight (**B**), and neonatal mortality (**C**). ((**A**) Proportion of twins at maternal ages 5–20 years, n = 203 breeding events; (**B**) mean cub birth weight at maternal ages 5–23 years, n = 369 births; error bars represent ± 1 SD using ANOVA with post-hoc Bonferroni corrections, *p* > 0.05; (**C**) proportion of neonatal mortalities at maternal ages 5–23 years, n = 476 births).

**Figure 3 animals-15-01182-f003:**
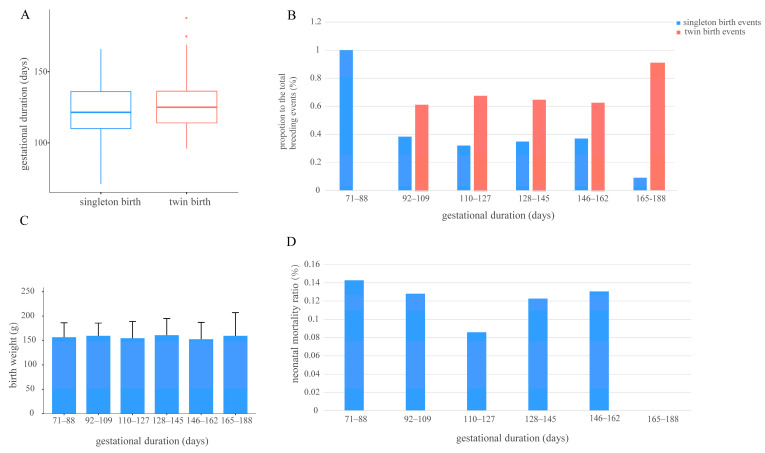
Median gestational duration for singletons and twins (**A**), distribution of singleton and twin births for different gestational durations as proportions (**B**), and relationships between gestational duration and cub birth weight (**C**), and between gestational duration and neonatal mortality (**D**). ((**A**) Gestational duration of singletons and twins, n = 324 breeding events; horizontal lines within the boxplot represent the upper quartile, median, and lower quartile, respectively; the whiskers represent minimum and maximum values within the range of non-abnormal data; the dots outside the box indicate outliers; (**B**) distribution of gestational durations for singletons and twins, n = 324 breeding events; (**C**) mean cub birth weights for six grouped gestational durations, n = 248 births; error bars represent ± 1 SD using ANOVA with post-hoc Bonferroni corrections, *p* > 0.05; (**D**) proportion of neonatal mortalities for six grouped gestational durations, n = 480 births).

**Figure 4 animals-15-01182-f004:**
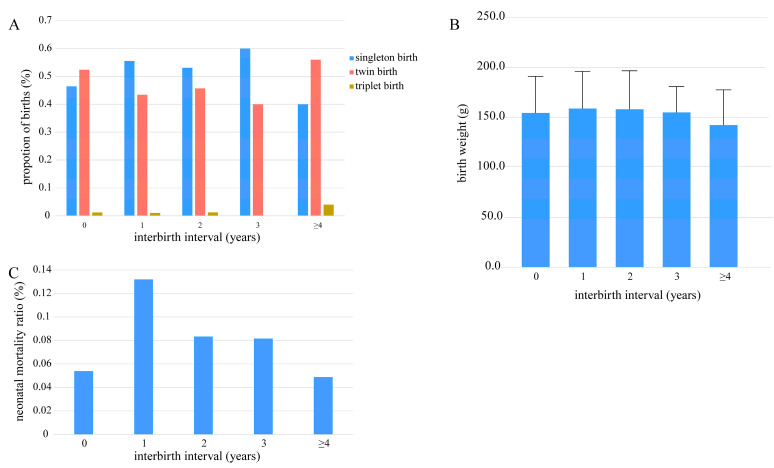
Interbirth intervals for singletons, twins, and triplets (**A**), relationships between interbirth intervals and cub birth weights (**B**), and distributions of neonatal mortality for different interbirth intervals (**C**). ((**A**) Interbirth intervals of singletons, twins, and triplets, n = 324 breeding events; (**B**) mean cub birth weights at five different interbirth intervals using ANOVA with post-hoc Bonferroni corrections, n = 444 births; error bars represent ± 1 SD, *p* > 0.05; (**C**) proportion of neonatal mortalities at five different interbirth intervals, n = 480 births. The “≥4” category includes intervals of 4 years or longer (n = 25 events). n = 324 breeding events).

**Figure 5 animals-15-01182-f005:**
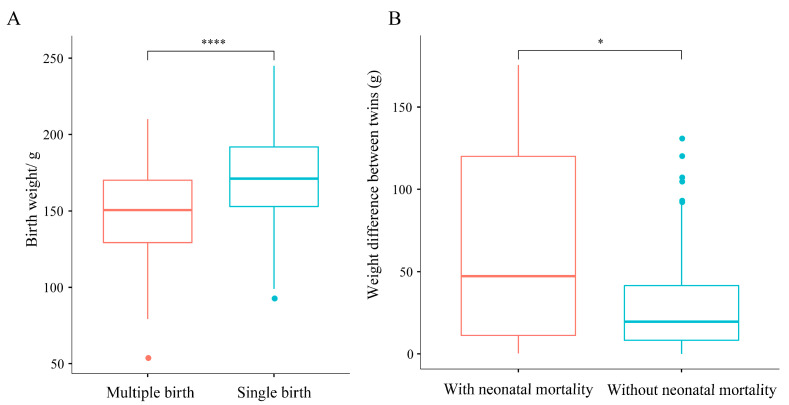
Cub weights for singleton and multiple births (**A**), and the difference between cub weights for twins with and without neonatal mortality (**B**). ((**A**) Cub weights for singleton and multiple births; horizontal lines within the boxplot represent the upper quartile, median, and lower quartile, respectively; the whiskers represent minimum and maximum values within the range of non-abnormal data; the dots outside the box indicate outliers. **** indicates a *p* < 0.0001 according to the Wilcoxon rank-sum test. n = 238 breeding events; (**B**) cub weights for twins with and without neonatal mortality; horizontal lines within the boxplot represent the upper quartile, median, and lower quartile, respectively; the whiskers represent minimum and maximum values within the range of non-abnormal data; the dots outside the box indicate outliers. * indicates a *p* < 0.05 according to the Wilcoxon rank-sum test. n = 133 breeding events).

## Data Availability

The data that support the findings of this study are available from the corresponding author upon reasonable request.

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
