# Peer review of "Exploring Captive Giant Panda Reproduction: Maternal and Offspring Factor Correlations from 324 Breeding Events"

_animals, 2025, doi:10.3390/ani15081182_

Round 1

Reviewer 1 Report

Comments and Suggestions for Authors

The manuscript offers to explore a large database of panda reproductive events to answer basic questions about the biology of reproduction in that species.  It is a much more comprehensive study compared to previous ones on the same topic.  However, because of the unique value of the analyzed data, authors are requested to provide more details and be more specific about their observations.

Here are some comments that have to be addressed:

- The methodology is too succinct.  The selection of breeding events is too vague and does not report enough information about the breeding history of the different females (besides the fact that they are 'healthy').

- Do all females have the same opportunities/chances to breed?

- Are all the breeding events comparable?  There is no information about natural mating or artificial insemination.  What about a possible paternal effect on the outcome? 

- Are the intervals between births comparable between females?

- What was the average weight of the females at the first pregnancy?

- What is the extent of individual maternal effects?  

- What was the maximum number of pregnancies per females?

- The analysis of the gestation length is not very informative because of the embryonic diapause that is variable.  This important fact is not even mentioned in the discussion.

- Overall, this is an interesting set of descriptive data.  However, authors have to be cautious about the analysis and interpretation of their findings.  For instance, this is not sure that breeding recommendations can be extrapolated.

Reviewer 2 Report

Comments and Suggestions for Authors

The manuscript describes records of births of captive giant pandas and interrogates the factors which predict successful reproduction in the species in the context of captive conservation. The data presented are valuable and have the potential to be informative for giant panda breeding specifically but potentially for other aspects of mammal conservation and breeding. The methodology requires elaboration and does not provide sufficient detail for the reader to be clear about precisely what has been done. Furthermore, in its current form, the presentation of the data is ambiguous and full of errors or inappropriate descriptions or visualisations. These issues undermine the value that the manuscript has to offer for the field. The authors should also be careful about the wording of the interpretation of their findings as the citing of a ‘framework’ is misleading and inappropriate; alternatively, the authors should present a framework that could be referred to. It is recommended that the authors revise the manuscript and resubmit for consideration.

Comments on the Quality of English Language

Some of the language used is misleading and could be improved through revision by an English first-language speaker.

Reviewer 3 Report

Comments and Suggestions for Authors

1. Brief Summary

The manuscript presents an analysis of maternal and gestational factors influencing cub survival and health in captive giant pandas. Utilizing a comprehensive dataset of 324 breeding events spanning from 1998 to 2023, the study identifies critical factors such as maternal age, interbirth intervals, and gestational duration that significantly affect cub birth weight and neonatal mortality. The primary contributions of this research lie in its thorough analysis, robust methodology, and the development of evidence-based breeding guidelines, which are essential for optimizing reproductive efficiency and supporting giant panda conservation efforts.

2. General Concept Comments

This reviewer commends the authors for their manuscript. The analysis of 324 breeding events represents one of the most extensive datasets published on the reproduction of captive giant pandas, allowing for more precise and reliable identification of patterns and trends. Moreover, the study effectively identifies maternal and gestational factors that significantly influence cub survival and health, such as maternal age, interbirth intervals, and gestational duration. The findings provide crucial insights for developing systematic breeding guidelines aimed at optimizing reproductive efficiency and supporting the recovery of the giant panda population.

While the article is of high quality, there are areas that may be enhanced:

  • The discussion does not address delayed embryonic implantation, a known phenomenon in ursids that can lead to significant variability in the interval between fertilization and implantation until birth. It would be beneficial to include a section discussing delayed embryonic implantation, elaborating on how this phenomenon might affect gestational duration in pandas and, consequently, the study's outcomes. Could this phenomenon impact the birth weight of the cubs? What is known about this in pandas? 

  • The article does not mention whether there were cases of pseudopregnancy among the pandas studied. This reviewer is curious to know whether this occurs in pandas and, if so, whether it was evaluated. Including such information, if possible, would enrich the manuscript.

  • The occurrence of split parturition, where cubs with the same parent are born on different days, reported in grizzly and American black bears, is not mentioned. This reviewer is unaware of its occurrence in pandas. Was this phenomenon observed in any cases? Has it ever occurred in pandas? If relevant, discussing this could add value to the manuscript.

Once again, this reviewer congratulates the authors on the important compilation of data.

Round 2

Reviewer 1 Report

Comments and Suggestions for Authors

Authors have properly addressed the comments.

Author Response

Comments 1: Authors have properly addressed the comments.

Response 1: Thank you for your positive assessment. We are pleased to know that our revisions adequately addressed your concerns. Your feedback has been invaluable in strengthening the methodological transparency of this work.

Reviewer 2 Report

Comments and Suggestions for Authors

The authors have made a good effort to address the comments raised on the previous version of the manuscript. In almost all instances, the authors have made satisfactory changes which have markedly improved the manuscript. However, one issue that remains problematic is the description of the statistical tests. As they are described, they are not reproducible. The tests should be individually described, clearly stating which variables were the response and which were the predictors in each case. In other words, each test should be detailed in a manner akin to the following example: 'In order to test the effect of maternal age and body condition on likelihood of a stillbirth, we used a generalised linear model using a Poisson distribution and a logit link function with percentage of stillbirths as the response variable, the maternal age as a continuous predictor and body condition score as a categorical predictor'. It is not enough to list all response variables together and all predictor variables together without explaining which were paired with which in the analyses. If the authors are concerned that it would make the text too long then this information should be provided as supplementary material. The fact remains that the methodology of the manuscript should be completely reproducible and in its current form, it is not.
